# Mitochondrial Autophagy in Ischemic Aged Livers

**DOI:** 10.3390/cells11244083

**Published:** 2022-12-16

**Authors:** Jae-Sung Kim, William C. Chapman, Yiing Lin

**Affiliations:** 1Department of Surgery, Washington University in St. Louis, St. Louis, MO 63110, USA; chapmanw@wustl.edu (W.C.C.); liny@wustl.edu (Y.L.); 2Department of Cell Biology & Physiology, Washington University in St. Louis, St. Louis, MO 63110, USA

**Keywords:** liver, ischemia/reperfusion, autophagy, mitochondria, mitochondrial dynamics

## Abstract

Mitochondrial autophagy (mitophagy) is a central catabolic event for mitochondrial quality control. Defective or insufficient mitophagy, thus, can result in mitochondrial dysfunction, and ultimately cell death. There is a strong causal relationship between ischemia/reperfusion (I/R) injury and mitochondrial dysfunction following liver resection and transplantation. Compared to young patients, elderly patients poorly tolerate I/R injury. Accumulation of abnormal mitochondria after I/R is more prominent in aged livers than in young counterparts. This review highlights how altered autophagy is mechanistically involved in age-dependent hypersensitivity to reperfusion injury.

## 1. Introduction

The liver has dual blood supplies. The second largest organ in the body is, thus, innately vulnerable to hypoxic and anoxic stress, especially in pericentral regions. Temporary blockade and subsequent resumption of hepatic blood supply during liver resection, transplantation surgery, cardiac arrest, and hemorrhagic shock inevitably induce ischemia/reperfusion (I/R) injury in the liver [1,2]. I/R injury is multifactorial, including the generation of reactive oxygen species (ROS) and nitrogen species, Ca^2+^ deregulation, loss of cellular antioxidants, stimulation of catabolic enzymes, reduced autophagy, and mitochondrial dysfunction [1,3,4]. Suboptimal livers, such as aged livers, steatotic livers, and donation after circulatory death (DCD) livers, are highly prone to I/R injury. These marginal livers are often considered to be unsuitable for transplantation because of their poor tolerance against I/R stress and greater risk for the recipient. Despite the advancement of organ preservation and surgical techniques, the shortage of donor livers remains a major obstacle in liver transplantation. In 2019, a total of 8896 liver transplants were performed in the United States (U.S.) with 13,093 patients on the waiting list [5]; it is alarming that one third of the patients awaiting life-saving liver transplantation never received donor livers. However, about 10% of procured donor livers are annually discarded in the U.S. due to concerns of graft failure [5]. If fully salvaged, these suboptimal livers would substantially expand the donor pool. It remains elusive how ischemia sensitizes suboptimal livers to reperfusion injury. Mitochondrial dysfunction and consequent energy failure are commonly observed after I/R.

Elderly patients poorly tolerate I/R injury [6]. Reduced reparative capacity with aging is causatively linked to decreased mitochondrial function after reperfusion [7,8,9]. Orchestrated with mitochondrial dynamics, the onset of mitochondrial autophagy (mitophagy) clears abnormal and dysfunctional mitochondria well before they harm neighboring healthy mitochondria. Impaired or insufficient autophagy, thus, results in hepatocyte death after I/R. This review describes our current understanding of the effects of aging on I/R injury in hepatocytes and highlights the mechanistic correlation of autophagy with the pathogenesis of age-dependent hypersensitivity to reperfusion injury. In addition, the controversies regarding post-transplant outcomes with aged donor livers are discussed.

## 2. Mitophagy and Mitochondrial Dynamics in Ischemic Aged Livers

### 2.1. Before Ischemia

Life expectancy at birth has substantially increased during the past century, leading to a marked growth in the senior population. It is anticipated that by 2030, 1 in 6 people around the globe will be over 60 years old [10], and the size of the elderly population will reach 20% of the total U.S. population [11]. As the incidence and severity of diseases and accidents elevate with aging, the number of elderly patients with hepatic malignancies who need surgical treatments is anticipated to prevail steadily. Noticeably, liver cancer is the fastest increasing cause of cancer death in the U.S. [12]. The liver changes morphologically and functionally with aging, including reduced liver weight, decreased metabolic capacities and hepatic blood flow [13,14,15]. Unlike the brain, heart, and muscle, the liver is, however, one of the least studied organs in the aging research community, since it is relatively resilient to the effects of aging. The liver carries out multiple integral functions, such as the synthesis of essential proteins and co-factors, storage of vitamins and minerals, secretion of hormones and bile, and detoxification of toxins and chemicals. Because these essential cellular tasks are heavily reliant on ATP, individual hepatocytes are loaded with 1000–1500 mitochondria [16]. In mice, the advancement of age from 8 months to 30 months decreases the number of mitochondria by 10% and 32% in the liver and the heart, respectively [17]. In addition, aged animals display a lower mitochondrial membrane potential [18,19], even though mitochondrial respiration through complex I and complex II remains intact [19]. This observation suggests the presence of an adaptive response of hepatic mitochondria to aging in order to counteract age-mediated alterations in mitochondrial oxidative phosphorylation. It has also been reported that mitochondrial oxygen uptake unexpectedly increases with advancing age [20]. Given that new mitochondria are continuously generated every 15 to 25 days in metabolically active cells, such as hepatocytes [21], a modest reduction in mitochondrial quantity may not necessarily accompany energy failure in the liver.

Autophagy is an evolutionarily conserved catabolic process that not only degrades unneeded or excessive intracellular proteins, but also digests dysfunctional organelles such as abnormal mitochondria [22]. Autophagy is a dynamic process [23]; after cytoplasmic constituents and abnormal mitochondria are partially enveloped by a membrane sac (phagophore), extension and closure of the phagophore occur, leading to the formation of a double-membrane structure called the autophagosome. Following the fusion between the autophagosome and the lysosome, acidic lysosomal enzymes hydrolyze cytoplasmic elements that are eventually recycled for other uses. So far, over 35 autophagy-related proteins (ATGs) have been identified. Selective mitochondrial autophagy (mitophagy) is a cellular event that is crucial to mitochondrial quality control. Timely onset of mitophagy clears abnormal mitochondria, thereby preventing the accumulation of defective or dysfunctional mitochondria in cells. Multiple proteins have been proposed to mediate mitophagy, including B-cell lymphoma 2 (Bcl2)/adenovirus E1B 19-kDa-interacting protein 3 (BNIP3) [24] [25], parkin/PTEN-induced putative kinase 1 (PINK1) [26], and FUN14 domain-containing protein 1 (FUNDC1) [27].

Three distinct types of mitophagy have been proposed [28]. The major difference between type I and II mitophagy is the involvement of phosphatidylinositol-3-kinase class III. The depolarization of mitochondrial membrane potential also determines the type of mitophagy. While type I mitophagy takes place in polarized mitochondria, type II mitophagy is triggered by mitochondrial depolarization. In polarized respiring mitochondria, PINK1 levels are quite low because the presenilin-associated rhomboid-like protein (PARL) rapidly cleaves PINK1 when it translocates into the mitochondrial intermembrane space [29]. Mitochondrial import of PINK1 is mediated by translocase of the inner membrane 23 (TIM23), translocase of the outer membrane 7 (TOM7) and overlapping with the m-AAA protease 1 (OMA1) [30]. When mitochondria lose their membrane potential in the presence of uncouplers, such as carbonyl cyanide m-chlorophenyl hydrazine (CCCP), TIM23-dependent mitochondrial import of PINK1 is prohibited, resulting in the accumulation of PINK1 in the mitochondrial outer membrane. Accumulated PINK1 subsequently recruits and activates the E3 ubiquitin ligase parkin from the cytosol to depolarized mitochondria. Mitochondrial outer membrane proteins, including mitofusin [31,32,33], voltage-dependent anion channel (VDAC) [34] and MITOL/March5 [35], become ubiquitinated by parkin thereafter. Aberrant mitochondria tagged with the ubiquitin moiety are later cleared by mitophagy [36]. Thus, PINK1 not only detects mitochondrial dysfunction, but also signals parkin to incite mitophagy. Type III mitophagy eliminates oxidized mitochondria through the formation of mitochondria-derived vesicles independently of mitochondrial membrane potential. Unlike type I and II mitophagy that enwraps entire mitochondria, type III mitophagy appears to remove only oxidized mitochondrial components [37].

We have reported that aging alone minimally impacts autophagy in the liver because basal and inducible autophagy in hepatocytes from 26-month-old mice are comparable to that from 3-month-old mice [7]. Chaperone-mediated autophagy declines in aged livers [38]. In contrast to mitophagy that selectively eliminates abnormal mitochondria, chaperon-mediated autophagy primarily clears cytosolic proteins.

Another cellular event accountable for sustaining mitochondrial quality is mitochondrial dynamics. Mitochondria are highly dynamic organelles that continuously fuse and divide. The balance between fusion and fission shapes mitochondrial morphology. Mitochondrial dynamics plays a cardinal role in lipid accumulation during aging, obesity, metabolic syndrome, and chronic liver disease [39,40,41]. In mammalian cells, fusion is governed by mitofusin (MFN) and optic atrophy 1 (OPA1) in mitochondrial outer and inner membranes, respectively, while fission is primarily mediated by dynamin-related protein 1 (DRP1) and fission-1-protein (FIS1) [42]. These opposing events not only determine mitochondrial shape and size, but also play a pivotal role in mitochondrial stress response [43]. It has been shown that parkin prevents the fusion between damaged and healthy mitochondrion, which promotes mitophagic clearance of damaged mitochondrion [44]. Parkin also boosts DRP1-dependent mitochondrial fission to facilitate mitophagy [45,46], confirming that mitochondrial dynamics is synergistically intertwined with mitophagy. Under stress, healthy mitochondria fuse or interconnect with injured mitochondria to compensate for defects in the injured organelles. This protective adaptation rescues 80% of the mitochondrial DNA defects by sharing individual mitochondrial contents [47,48]. Fusion can also take place between two damaged mitochondria by cross-complementation to one another in order to mitigate mitochondrial damage through the exchange of proteins and lipids with another mitochondrion [49]. The importance of fusion in cell survival is further corroborated by observations that the strategies that promote mitochondrial interconnection confer cytoprotection against nutrient insufficiency [50], oxidative stress [51], and cardiac and renal reperfusion injury [52,53]. Mitochondrial fusion has, thus, been proposed to serve as a first line of defense against acute stress [54]. Prior to ischemia, basal levels of MFN1, MFN2, and DRP1 in aged hepatocytes are similar to those in young hepatocytes [8]. Furthermore, nearly all mitochondria are oval or round in shape in both young and aged cells, suggesting that little to no alterations in mitochondrial dynamics occurs in aged normal livers. Accumulating evidence indicates that OPA1 is a critical regulator of mitochondrial dynamics [55,56,57]. Under stress, OMA1, a mitochondrial metalloprotease, cleaves the S1 site of long (L)-OPA1 to generate short (S)-OPA1 [56]. While L-OPA1 induces fusion, S-OPA1 promotes fission [57]. Similar to MFN1/2 and DRP1, basal levels of L- and S-OPA1 are comparable between young and aged cells [9]. Taken together, prior to ischemia, aged livers sustain mitophagy and mitochondrial dynamics.

### 2.2. During Ischemia

During ischemia, the cessation of blood flow causes tissue anoxia and nutrient depletion. Concurrently, anaerobic glycolysis and hydrolysis of ATP rapidly decrease cellular pH [58]. These ischemic changes depolarize mitochondrial membrane potential and decrease the pH gradient across mitochondrial inner membranes, leading to the loss of proton motive force. While prolonged ischemia irreversibly damages mitochondria through alterations in the electron transport chain and destabilization of mitochondrial proteins, a short duration of ischemia does not. Upon reperfusion following low-to-moderate ischemia, mitochondria rapidly repolarize and resume ATP generation [7,59]. The mitochondrial matrix under physiological conditions is alkaline. Noticeably, pH considerably affects ROS production. In isolated mitochondria incubated at different pH, the rate of ROS formation substantially declines at acidic pH [60], suggesting cytoprotective or beneficial effects of ischemic acidosis. Indeed, acidosis strongly protects hepatocytes from reperfusion injury [59], chemical hypoxia [61], and drug-induced hepatotoxicity [62]. In ischemic livers, the recovery of normal pH upon reperfusion is an independent factor that culminates in I/R injury [59]. When ischemic livers are reoxygenated under acidic conditions, hepatic I/R injury is substantially subdued. Cytoprotection by acidic pH is, at least in part, mediated by the deactivation of degradative enzymes, including phospholipases and proteases, many of which become fully activated at neutral or weakly alkaline pH [63,64]. Importantly, intracellular acidosis blocks the onset of mitochondrial permeability transition (MPT) [65]. Unlike mitochondrial outer membranes that are permeable to solutes with a low molecular weight, mitochondrial inner membranes are virtually impermeable to all solutes, except through specific transporters. However, pathological conditions that elevate mitochondrial Ca^2+^ and ROS open the high-conductance permeability transition (PT) pores in mitochondrial inner membranes, leading to MPT onset and non-selective diffusion of solutes up to 1500 Da [4]. As a consequence, oxygen uptake in mitochondria becomes uncoupled from ATP production. This futile cycle without ATP generation eventually depletes cells of ATP, resulting in necrotic cell death [4]. The onset of MPT also releases pro-apoptotic proteins such as cytochrome c that are normally sequestered in the intermembrane space. Subsequent stimulation of caspases induces apoptotic cell death [66]. The presence or absence of glycolytic ATP governs the mode of cell death after I/R. The structural and molecular identity of PT pores remains elusive. It has long been postulated that PT pores consist of the anion nucleotide translocator in the inner membrane, VDAC in the outer membrane, and cyclophilin D from the mitochondrial matrix [67]. However, recent studies suggest the involvement of F-ATP synthase that resides in inner membranes [68,69].

The citric acid cycle intermediate, succinate, can accumulate during ischemia as a consequence of reversed succinate dehydrogenase [70]. Although succinate accumulation is observed in ischemic hearts across species, it may not be a universal event for all organs [71]. It has been proposed that ischemic accumulation of succinate is, indeed, a protective event in the heart [72].

During short ischemia, both young and aged cells remain viable due to a lack of MPT onset [7]. Nutrient depletion or starvation is a powerful stimulus of autophagy. Although livers are subjected to nutrient depletion during ischemia, concurrent depletion of ATP certainly suppresses the onset of autophagy, as the execution of autophagy is a highly energy-consuming process [73,74]. During ischemia, hepatocellular ATP decreases to 0.6% of normoxic values [59]. The sequestration step and the lysosome-dependent degradation step are, in particular, ATP-sensitive [74]. Noticeably, ischemia substantially decreases the expression of many autophagy-related proteins, such as ATG4B, ATG12-5, BECN1, and ATG5, in both young and aged hepatocytes [7,9]. The decrease in ATGs during ischemia is likely to be mediated by calpains, Ca^2+^-dependent proteases, because pharmacological and genetic inhibition of calpains delays this ischemia-dependent loss of ATGs [7,75]. During ischemia, cytosolic Ca^2+^ increases as a result of ATP depletion, the collapse of Na^+^ and K^+^ gradients and inhibition of ATP-dependent Ca^2+^ pumps [76], which in turn stimulates calpains [7,77]. Immunoblot analysis of calpain activity revealed that calpain stimulation is more prominent in aged hepatocytes [7]. The sensitivity of ATGs to calpains has been well documented [78,79,80]. In particular, calpain-mediated degradation of ATG5 not only affects autophagy, but also sensitizes cells to apoptosis [79]. Calpains cleave full-length ATG5 (~33 kDa) at Thr193 into short-length ATG5 (~24 kDa). This truncated ATG5 can translocate from the cytosol to mitochondria and interact with B-cell leukemia-X long (Bcl-XL), causing the permeabilization of mitochondrial outer membranes and mitochondrial depolarization [79,81]. A considerable increase in truncated ATG5 has been observed in ischemic aged hepatocytes [9]. Nonetheless, because both autophagy and apoptosis rely on cellular ATP, anoxic depletion of ATP virtually halts both events.

While the proteins involved in mitochondrial dynamics remain unchanged in normoxic aged hepatocytes, ischemia noticeably decreases the expression of MFN1, MFN2, and DRP1 [8,9]. The reduction in these mitochondrial outer membrane proteins is, at least in part, caused by calpain activation. Assessment of the stability of MFN in the presence of cycloheximide, a protein synthesis blocker, demonstrates that its half-life time (t _1/2_) is much shorter in aged hepatocytes than in young cells. Moreover, intracellular acidification during ischemia further destabilizes this fusion protein in aged cells, indicating that mitochondrial dynamics in aged livers is altered during ischemia [8]. Collectively, during short ischemia, aged livers lose some key proteins that are essential to mitophagy and mitochondrial dynamics. Nonetheless, PT pores remain closed. Consequently, cell death in aged livers is minimal.

### 2.3. After Ischemia

Reperfusion restores pH, and resumes the delivery of oxygen and nutrients to cells and tissues. During the early phase of reperfusion, mitochondria reestablish the proton motive force to produce ATP, leading to the initiation of mitophagy. Noticeably, the mitochondrion engulfed in the autophagosome remains polarized at this time, suggesting that mitophagy onset precedes mitochondrial depolarization and that depolarization-induced type II mitophagy may not occur in reperfused hepatocytes [78]. However, such a recovery of autophagy is temporary and autophagic flux completely disappears well before cells undergo necrosis.

While ischemia elevates predominantly cytosolic Ca^2+^ without altering mitochondrial Ca^2+^, reperfusion progressively raises mitochondrial Ca^2+^, a causative event that instigates MPT onset [76,82]. Repolarization of mitochondria following reperfusion drives mitochondrial Ca^2+^ uptake through the electrogenic mitochondrial Ca^2+^ uniporter. Mitochondrial Ca^2+^ overloading is sufficient to induce MPT, which is further potentiated by other factors, such as the restoration of neutral pH [82]. In the liver, Ca^2+^-loaded mitochondria do not undergo MPT at a pH below 6.4, but begin to develop a large amplitude of swelling from pH 6.8 [83]. Half-maximal opening of PT pores in the presence of Ca^2+^ takes place at pH 7.0. The recovery of normal pH upon reperfusion removes the inhibitory effects of acidic pH on MPT. Ratiometric assessment of pH in rat hepatocytes demonstrates the progressive normalization of intracellular pH during reoxygenation at pH 7.4 following ischemia at pH 6.2, with half-maximal recovery at 30 min after reperfusion [59]. Hence, a lethal combination of mitochondrial Ca^2+^ overloading and restoration of normal pH rapidly provokes MPT onset. As a result, ROS- and Ca2+-laden mitochondria lose their permeability barrier and fail to produce ATP, leading to cell death. However, the aforementioned deleterious events rarely occur to young hepatocytes, as mitophagy clears abnormal mitochondria well before the pervasive onset of MPT. Robust autophagic flux manifests in young hepatocytes after short I/R [7,8,9]. Of note, the mitochondrial population in hepatocytes is heterogeneous; a discrete pool of mitochondria is more vulnerable to stress, rapidly undergoes MPT, and releases Ca^2+^ and ROS to neighboring healthy mitochondria, which eventually causes widespread MPT [82,84]. In agreement with this view, we have repeatedly observed that the widespread onset of MPT in reperfused hepatocytes is preceded by the initial MPT in a small number of mitochondria [78], suggesting that prompt removal of a discrete pool of stress-prone mitochondria is indispensable for cell survival after I/R. Early blockade of ROS, therefore, prevents this extensive MPT onset and cell death after reperfusion. Hepatocyte injury during the early phase of reperfusion instigates systemic inflammatory injury during the late phase of reperfusion [85]. Stimulation of autophagy reduces inflammation, as well as cell death, in ischemic livers [86,87,88,89], substantiating that sustaining autophagy in hepatocytes is critical for liver viability throughout reperfusion.

Before I/R, baseline levels of ATGs and basal autophagic flux are indistinguishable between young and aged cells. As is the case with young hepatocytes, hepatocytes from aged mice have a strong autophagic flux under normoxia and starvation, which is indicative of the intrinsic resilience of hepatic autophagy during aging. However, a striking reduction in autophagy occurs when aged livers are exposed to I/R. Unlike young hepatocytes, aged cells fail to execute mitophagy during reperfusion. As a result, they accumulate abnormal mitochondria and generate uncontrolled ROS, all of which result in energy failure, and ultimately hepatocyte death [7]. In a model of simulated I/R using isolated hepatocytes and in vivo livers from young and aged mice, I/R rapidly depletes aged livers of ATG4B in a calpain-dependent manner. After 120 min of reperfusion, ATG4B expression in aged hepatocytes decreases to 20% of normal values [7]. Overexpression of ATG4B, however, reverses defective autophagy, MPT and cell death after I/R, denoting a causal role of ATG4B loss in age-dependent I/R injury. ATG4B is a redox-sensitive cysteine protease that plays an integral role in the recycling of microtubule-associated proteins 1A/1B light chain 3B (LC3) [90]. BECN1 also confers cytoprotection against heightened I/R injury in aged livers, as BECN1 overexpression suppresses reperfusion-induced ATG4B loss [7]. BECN1 is a multi-functional protein involved in autophagy initiation and cell survival [91,92]. BECN1-dependent protection is likely to be associated with its physical interaction with ATG3, a ubiquitin-like conjugating enzyme in the LC3 recycling process [90].

Autophagy is mechanistically intertwined with various cellular survival pathways [93]. Growing evidence indicates that a post-translational modification of mitochondrial proteins, such as acetylation/deacetylation, directly influences mitophagy [94]. In ischemic livers, the viability of hepatocytes is highly dependent on the status of acetylation/deacetylation of mitofusin 2 (MFN2), a mitochondrial fusion protein located at mitochondrial outer membranes [8,95]. Compared to young livers, the acetylated form of MFN2 is prevalent in aged livers. Approximately one third of mitochondrial proteins are acetylated, 24% of which are closely linked to energy homeostasis [96]. Covalent addition of an acetyl group to lysine residues (by lysine acetyltransferases) and removal of the acetyl group from lysine (by deacetylases) is a dynamic process that affects a variety of cellular functions, such as DNA binding affinity, catalytic activity, stability and localization of target proteins [97]. Hyperacetylation of mitochondrial proteins is salient in steatotic and alcoholic livers [98,99]. Sirtuins are the mammalian ortholog of yeast silent information regulator 2 (Sir2) and can deacetylate numerous molecular targets, using oxidized nicotinamide adenine dinucleotide (NAD^+^) [100,101,102]. Among seven different isoforms of sirtuins in mammals, cytosolic sirtuin 1 (SIRT1) deacetylates MFN2 in the liver and this SIRT1-MFN2 axis plays a pivotal role in hepatic I/R injury [8,95]. Genetic and biochemical approaches using deletion and point mutants reveal that SIRT1 enhances autophagy through deacetylating C-terminal lysine residues (K655 and K662) in MFN2 [95]. While basal levels of SIRT1 and MFN2 are comparable between young and aged hepatocytes, I/R drastically reduces their expression in aged cells [8]. Within 1 h of ischemia, aged hepatocytes lose 80% of SIRT1 from basal values, whereas young hepatocytes show only a 10% reduction. Reperfusion of aged cells further diminishes SIRT1 expression to nearly undetectable levels. Similar to SIRT1, MFN2 in aged cells is nearly completely depleted after I/R. Multiple factors contribute to the loss of SIRT1 and MFN2 during I/R, including intrinsic protein instability, cellular acidosis during ischemia, and calpain activation [8]. MFN1, another isoform of MFN, is also lost in reperfused aged cells. Paradoxically, overexpression of MFN1 aggravates cell death after I/R. In ischemic hepatocytes, we have reported that MFN2, but not MFN1, is a substrate for SIRT1 [95]. MFN2 may be a multifunctional protein. Besides its role in facilitating mitochondrial fusion, MFN2 regulates the tethering between mitochondria and the endoplasmic reticulum [103,104] and the fusion of autophagosomes with lysosomes [105]. It is noteworthy that basal levels of NAD^+^ and NAD^+^/NADH in aged cells are markedly lower than those in young cells [8], insinuating that even though livers are able to sustain SIRT1 expression with advancing age, SIRT1-dependent deacetylation inherently declines in aged livers. Either the decrease in NAD^+^ or increase in NADH (reduced NAD+/NADH) in mitochondria accompanies the reduction in mitochondrial respiration, electron transport chain activity, and ATP generation [106]. Intriguingly, overexpression of either SIRT1 or MFN2 alone fails to prevent reperfusion-induced injury in aged livers. Instead, dual overexpression of MFN2 and SIRT1 promotes mitophagy, prevents the onset of MPT, and improves hepatocyte viability after reperfusion, underscoring the importance of the SIRT1–MFN2–mitophagy signaling axis in the survival of aged livers after I/R.

Aged hepatocytes are depleted of key mitochondria dynamics-related proteins, such as MFN1, MFN2, DRP1, and L-OPA1, within 20 min of reperfusion [9]. However, confocal microscopy of mitochondria in live hepatocytes exhibits a normal round or oval shape of mitochondria in the aged group. As the balance between fusion and fission governs mitochondrial shape, the decline in both fusion and fission factors may render aged cells unable to sustain the overall balance of mitochondrial dynamics. It is, however, worth noting that the short form of OPA1 (S-OPA1), a fission inducer, progressively increases in reperfused aged cells, whereas the long form of OPA1 (L-OPA1), a fusion inducer, decreases [9], suggesting a shift in mitochondrial dynamics toward fission during reperfusion. The balance of L- and S-OPA1 is critical to cell survival and excessive processing of OPA1 causes tissue injury [107]. Mitochondria have two inner membrane proteases that regulate OPA1 processing, OMA1 and yeast mitochondrial DNA escape 1-like (YME1L) [108]. OMA1 is synthesized as a 60 kDa pre-proprotein that proteolytically matures into a ~40 kDa form. The smaller form of OMA1 is known to function as a stress sensor that is responsible for OPA1 cleavage [108,109]. Mitochondrial depolarization stimulates OMA1 independently of its protease activity [108]. Thus, loss of mitochondrial membrane potential after reperfusion in aged cells could favor OMA1 activation and subsequent cleavage of OPA1. This truncated form of OPA1 induces cell death, whereas YME1L-mediated cleavage of OPA1 promotes mitochondrial ATP generation [110]. Nonetheless, OPA1 cleavage by YME1L is less likely to occur during I/R because OMA1 promptly degrades YME1L under low ATP conditions, such as ischemia [111]. It has also been shown that YME1L becomes inactive in depolarized mitochondria [108]. Accumulation of S-OPA1 may contribute to a shift in mitochondrial dynamics toward fission in reperfused aged hepatocytes.

Mitochondrial dynamics and mitophagy work together to provide maximal quality control of mitochondria [54]. When a dysfunctional mitochondrion fails to fuse with a healthy mitochondrion, the former preferentially undergoes fission to facilitate mitophagic clearance [46]. Mitophagosomes engulf smaller or fragmented mitochondrion more readily [26,46,112]. Nevertheless, owing to a lack of cellular energy and loss of ATG4B, MFN2 and SIRT1, aged cells fail to execute mitophagy.

The loss of ATGs, MFN2, and SIRT1 is post-transcriptional, since their mRNA levels remain unchanged during I/R [8,9]. Ca^2+^ overloading and subsequent stimulation of calpains is an important cellular event that culminates in age-dependent I/R injury. Under non-pathological conditions, calpains are tightly regulated by calpastatin (CAST), an endogenous calpain-specific inhibitor [113]. CAST is nearly completely in a random coil conformation that consists of XL and L domains, followed by four identical inhibitory domains [114]. This natural inhibitor has a unique structural flexibility that not only facilitates its binding to calpains, but also escapes proteolysis by calpains by looping out and around its active site cysteine [115,116]. One CAST molecule blocks four calpain molecules [114], which is indicative of its robust inhibitory efficacy. X-ray crystallography studies show that CAST inhibits calpains by binding to two penta-EF domains and one catalytic cleft of calpains [115,116]. While CAST knockout (KO) mice display normal phenotypes and physiology at birth [117,118], they develop tissue damage upon exposure to stress, signifying an integral role of CAST in stress responses. In human livers and mouse hepatocytes and livers, reperfusion significantly decreases CAST expression in the aged group, but not in the young group [9]. Unexpectedly, compared to the young group, a greater expression of basal CAST was found in the aged group. However, CAST in aged cells is intrinsically unstable and short-lived even before ischemia and is rapidly lost during I/R. Adenoviral overexpression of CAST in aged mouse hepatocytes and livers prevents the onset of MPT, mitochondrial depolarization and necrotic cell death after I/R, while promoting autophagy and mitochondrial fusion [9]. CAST overexpression also increases the expression of BECN-1, ATG7, and ATG12-5, but decreases the formation of truncated ATG5, an injurious fragmented form of ATG5. Noticeably, CAST overexpression suppresses the conversion of L-OPA1 into S-OPA1, concomitantly with mitochondrial elongation in aged hepatocytes. Overall, calpain inhibition through CAST enhances both autophagy and mitochondrial fusion, two cellular events that are imperative to cell survival under stress conditions.

## 3. Controversy about Age Factor in Liver Surgery of the Elderly

Liver tumors occur largely in elderly patients (>65 years of age) [119] and resection surgery is the only curative treatment. Resection appears to be as beneficial to the elderly as it is for younger patients [120]. Despite the advancement of surgical techniques and specialized perioperative care, higher mortality rates following liver resection have been documented in elderly patients [121,122,123]. Numerous animal models firmly support that aged livers have a significantly lower reparative capacity following surgical stress associated with liver resection and transplantation. However, some studies do not support the causative roles of age in poor surgical outcomes [119,124,125]. One meta-analysis may explain this discrepancy, as it was reported that elderly patients with colorectal metastatic cancer are 2.7-fold more susceptible to I/R injury than their younger counterparts [126], suggesting that mortality and morbidity after resection are highly dependent on the type of cancer. It should also be noted that the intraoperative procedures performed on young and aged patients may not be identical, with a greater frequency of anatomic resections performed on younger patients as compared to wedge resections in the elderly [126]. Scrupulous selection of patients to avoid heightened perioperative risk may be another reason behind the controversy [127]. Moreover, nutritional balance, residual liver function, and muscle mass, all of which decline with aging, are also strongly correlated with surgical outcomes [128]. A single center, retrospective study with 332 patients that underwent curative anatomic resection showed that patients aged 75 years and above had a substantially increased risk of developing liver failure and likely had a longer length of stay in the hospital and greater inpatient cost [129].

Liver transplantation is the only effective remedy for patients with end-stage liver disease. Despite the increasing use of liver grafts from elderly donors, the discard rate for aged livers is still higher than that for younger livers [130]. In the U.S., about 13% of donor livers at the age of 55 and above were not transplanted in 2020. While studies using rodent models solidly support that age is an independent factor that worsens reperfusion injury during transplantation [131,132,133], issues have arisen in recent years as to whether age is truly a limiting factor in determining the suitability of donor livers [134,135,136]. Although there is no consensus on the upper age limit for donor grafts, it is noteworthy to mention two important factors that lead to substantial improvement in transplant outcomes with aged donor livers. First, cold ischemia exposed to liver grafts significantly decreased from 10 h (1990–1994) to 6 h (2010–2014) [137]. Adverse impacts of cold ischemia on graft survival are more prominent in aged livers than in young livers [138,139]. As such, transplant surgeons often perform the recipient’s hepatectomy before the arrival of the aged graft in order to reduce cold ischemic time. Second, the optimization of donor–recipient matching is a common practice nowadays. This improved selection of recipients markedly increases the success of transplantation with aged donors [137]. Preferred recipients who are less likely to incur additional risks associated with aged donor liver include the following: (1) first-time transplant recipients over the age of 45 years, (2) a body mass index (BMI) < 35, and (3) a cold ischemic time < 8 h [140]. Other recipient factors, such as mechanical ventilation, portal thrombus, and low serum sodium value, appear to also be important in the survival of aged grafts [141]. Interestingly, the age of recipients may not be a major risk factor [141,142,143].

## 4. Conclusions and Future Perspectives

Despite the increasing utilization of aged donor livers in the past few decades, many aged liver grafts are discarded. Aged livers are more vulnerable to reperfusion-induced mitochondrial stress than younger ones (Figure 1). Mitochondrial dynamics and mitophagy are two major events that are essential to sustaining the quality of mitochondria. Pharmacological or genetic modulations that enhance these events could lead to the development of new strategies, reducing I/R injury in elderly patients. Our understanding of mechanistic connections between mitochondrial dynamics and mitophagy in reperfused aged livers remains limited. One challenge associated with determining the alterations in mitochondrial dynamics is its complicated nature. As mitochondria can undergo partial fusion as well as full fusion [144,145], microscopic analysis of morphology alone may not accurately assess mitochondrial dynamics [146]. The recent development of mitochondria-targeted photoactivatable fluorophores [147,148] and mitochondria-specific liposomes [149] will enrich our knowledge of mitochondrial dynamics. In addition, future studies are warranted to elucidate how aging and I/R affect mitochondrial quality control in other hepatic cells, including Kupffer cells, endothelial cells, stellate cells, and cholangiocytes. In conclusion, mitochondrial dysfunction that results from altered mitochondrial dynamics and defective mitophagy is a key pathological mechanism underlying age-dependent hypersensitivity to I/R injury.

During normoxia, both young and aged hepatocytes sustain mitophagy and the balance between mitochondrial fusion and fission. During ischemia, hepatocytes encounter anoxia, ATP loss, and tissue acidosis. Short ischemia minimally affects mitochondrial dynamics in both young and aged cells. Autophagy, a highly energy-dependent cellular event, becomes inactive due to a lack of ATP during ischemia. At this time, ischemic acidosis blocks the onset of MPT. Reperfusion restores normal pH levels and resumes oxygen delivery. Young hepatocytes during reperfusion may sustain the dynamic balance between mitochondrial fusion and fission. In addition, active mitophagy can clear abnormal mitochondria that escape from mitochondrial fusion. Hence, neither MPT onset nor hepatocyte death occurs to reperfused young hepatocytes. However, reperfusion of aged hepatocytes converts L-OPA1 into S-OPA1. Loss of CAST activates calpains, leading to degradation of MFN2. These events shift mitochondrial dynamics toward fission. Calpain activation also depletes ATG4B and SIRT1. Consequently, aged hepatocytes after I/R accumulate abnormal mitochondria, which in turn causes the widespread onset of MPT, ATP depletion, and ultimately death.

## Figures and Tables

**Figure 1 cells-11-04083-f001:**
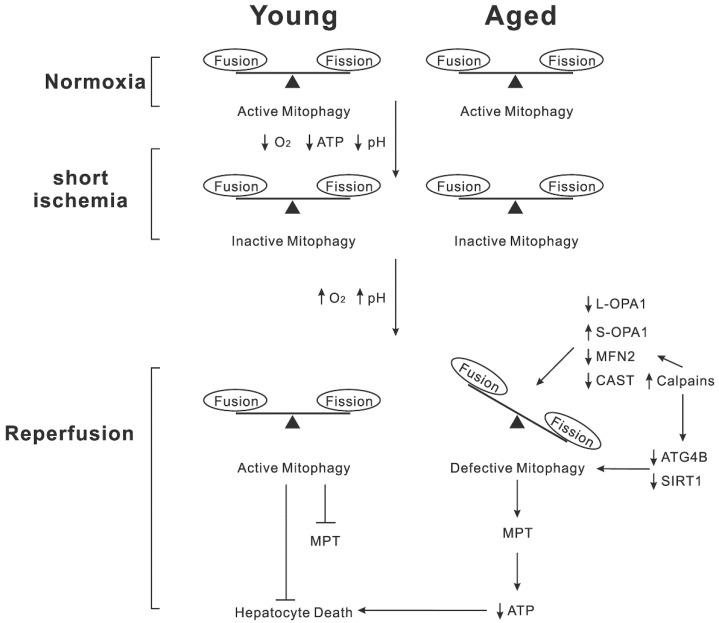
Proposed mechanism of aged hepatocyte death after I/R.

## Data Availability

Not applicable.

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
