# Peer review of "Mitochondrial Autophagy in Ischemic Aged Livers"

_cells, 2022, doi:10.3390/cells11244083_

Round 1

Reviewer 1 Report

The review by the Dr. Kim et al. discusses the involvement of mitochondrial quality control (MQC) in age-dependent hypersensitivity to reperfusion injury.

According to the title, the manuscript is not very well organized and structured, and specific sections require a major focus on MQC. Introduction and section 2a could be restructured with a common underlying theme. Also, though the only one figure presented is quite clear its quality is not excellent. The literature quoted is not exhaustive and updated.

Specific comments will be detailed below: 

1)    Introduction: the introduction requires a clear structure focused on MQC and should be improved. The author may apply the following structure:

a.     types of liver ischemia-reperfusion injury (IRI)

b.     what is mitochondrial quality control (MQC)

c.     which processes are involved in MQC (e.g. ROS scavenging, chaperones and proteolytic enzymes, the ubiquitin-proteasome system (UPS), the mitochondria-specific unfolded protein response (UPRmt), proteostasis, biogenesis, dynamics, mitophagy)

d.     translational significance of the MQC (i.e., relevance for liver ischemia-reperfusion injury)

2)    Section 2a. Before ischemia: this section should be restructured. In particular, the description of mitophagy is not sufficient and requires a brief description of the pathways (e.g. PINK1/Parkin axis and Parkin-independent pathways).

3)    Section 2b. During ischemia: please describe the role of mitochondrial succinate accumulation in IRI damage

4)    Section 2b. After ischemia: please describe the role of PINK1-mediated mitophagy in hepatic I/R-induced mitochondrial damage and inflammation.

Please, in lines 214-215 (…well-timed enactment of mitophagy is critical to cell survival after I/R….) specify the concept and add references.

5)    Section 3: I am not sure that this section is necessary. Alternatively, it may be reduced

6)    Conclusion and future perspectives: please add references (line 399….. connection between mitochondrial dynamics and mitophagy….)

Author Response

Responses to Reviewer 1

  1. Introduction: the introduction requires a clear structure focused on MQC and should be improved…. what is mitochondrial quality control (MQC)….. which processes are involved in MQC (e.g. ROS scavenging, chaperones and proteolytic enzymes, the ubiquitin-proteasome system (UPS), the mitochondria-specific unfolded protein response (UPRmt), proteostasis, biogenesis, dynamics, mitophagy)……translational significance of the MQC (i.e., relevance for liver ischemia-reperfusion injury)”

            We agree with this reviewer’s comments. Since this manuscript exclusively focuses on autophagy and mitochondrial dynamics, we have removed “mitochondrial quality control” from the manuscript. We have also changed the title to “Autophagy in ischemic aged livers” and revised the abstract and the introduction on pages 1 and 2. All changes are denoted in red in the revised manuscript.

  1. “Section 2a. Before ischemia: this section should be restructured. In particular, the description of mitophagy is not sufficient and requires a brief description of the pathways (e.g. PINK1/Parkin axis and Parkin-independent pathways)”

            As suggested, on page 3 in the revised manuscript, we have described different types of mitophagy.  

  1. “Section 2b. During ischemia: pleasedescribe the role of mitochondrial succinate accumulation in IRI damage”

            To our knowledge, succinate accumulation during ischemia is tissue-specific. Furthermore, this cardiac event may not be harmful during ischemia. On page 5 in the revised manuscript, we have described this.

  1. “Section 2b. After ischemia: pleasedescribe the role of PINK1-mediated mitophagy in hepatic I/R-induced mitochondrial damage and inflammation”

            As suggested, a revision has been made on page 6.  

  1. “Please, in lines 214-215 (…well-timed enactment of mitophagy is critical to cell survival after I/R….) specify the concept and add references”

            We have removed this sentence in the revised manuscript.

  1. “Section 3: I am not sure that this section is necessary”

            The goal of this manuscript is to validate and translate data from animal studies into human livers in clinical settings. The concept of age-dependent I/R injury has recently been challenged by clinicians and clinician-scientists. Therefore, there exist controversies as to whether aging adversely impacts surgical outcomes after transplantation, which is an important issue that we should not overlook. We believe that this section is needed to explain the discrepancies between mouse and human data.    

  1. “Conclusion and future perspectives to add references”

            We have revised and added references on page 10 of the manuscript.   

Reviewer 2 Report

Major comments:

1.       The review needs significant reorganization in terms of sub-headings and can be broken up into more scientifically distinct concepts.

2.       The role of proteins such as PARKIN in mitochondrial dynamics has not been discussed

3.       Details regarding current methods to quantify mitochondrial dynamics in vivo in the liver, their co-analysis with mitophagy need to be discussed including the strengths and limitations of each technique. These details are required to guide readers to understand the strengths of all discoveries and validity of results published in the review.

4.       The impact of altered dynamics on metabolism needs special attention

5.       How does the ATP production relate with NAD/NADH levels and electron transport chain activity?

6.       Future perspectives section needs an additional paragraph on broader discussion of the field and therapeutic targeting strategies/opportunities/challenges.

7.       The illustration or diagram is very superficial in its current form. Details regarding the major proteins involved and some dynamic changes under all the physiological conditions discussed should be illustrated more clearly.

8.       It is unclear whether authors are discussing hepatocyte specific changes or phenomenon in all cells of the liver. This should be clarified early in the review

9.       The role of antioxidants within and outside the mitochondria are central to the concept of ischemia reperfusion and need to be addressed in more detail.

10.   Overall more subheadings, breakdown of concepts and clear distinction of dynamics and mitophagy related pathways need to be delineated.

Author Response

Responses to Reviewer 2

  1. “The review needs significant reorganization in terms of sub-headings and can be broken up into more scientifically distinct concepts…..The role of proteins such as PARKIN in mitochondrial dynamics (MD) has not been discussed”

            We are very grateful to this reviewer for insightful suggestions and comments. We have revised the manuscript substantially. Parkin in MD has been described on page 3 in the revised manuscript. All changes are denoted in red in the revised manuscript.

  1. “Details regarding current methods to quantify MD in vivo in the liver, their co-analysis with mitophagy need to be discussed including the strengths and limitations of each technique. These details are required to guide readers to understand the strengths of all discoveries and validity of results published in the review.”

            On page 10 of the revised manuscript, we have added this important information.

  1. “The impact of MD on metabolism needs special attention”

            Though we well acknowledge the importance of MD in hepatic metabolism, the current manuscript focuses on acute I/R injury. We respectfully believe that details on MD-mediated metabolism in the liver is beyond scope of this manuscript. We have briefly described this on page 3 in the revised manuscript.

  1. “How does the ATP production relate with NAD/NADH levels and electron transport chain activity?”

            On page 7 of the revised manuscript, we have added this information.

  1. “Future perspectives section needs an additional paragraph on broader discussion of the field and therapeutic targeting strategies/opportunities/challenges.”

            We are very grateful for this comment. As suggested, we have revised the manuscript on page 10.

  1. “The illustration or diagram is very superficial in its current form. Details regarding the major proteins involved and some dynamic changes under all the physiological conditions discussed should be illustrated more clearly.”

            As suggested, we have added key players in the diagram of Figure 1.

  1. “It is unclear whether authors are discussing hepatocyte specific changes or phenomenon in all cells of the liver. This should be clarified early in the review”

            We have revised the manuscript (page 2) to clarify that the current manuscript largely covers hepatocytes. In addition, we have replaced “livers” with “hepatocytes” throughout the manuscript unless in vivo data are available. 

  1. “The role of antioxidants within and outside the mitochondria are central to the concept of ischemia reperfusion and need to be addressed in more detail.”

            The beneficial effects of antioxidants have been added on page 6 in the revised manuscript.

  1. “Overall more subheadings, breakdown of concepts and clear distinction of dynamics and mitophagy related pathways need to be delineated.”

            We have revised the title, abstract, introduction, main text, and diagram. Changes are denoted in red.

Round 2

Reviewer 1 Report

This review has improved and I am overall satisfied with the corrections. However, the first author already published a review titled “Autophagy in ischemic aged livers” as follow:

Choonghee Lee, Jae-Sung Kim,

Autophagy in ischemic aged livers,

Liver Research,

Volume 2, Issue 3,

2018, Pages 133-137,

ISSN 2542-5684,

https://doi.org/10.1016/j.livres.2018.09.006.

Therefore, title should be modified also including an aspect of mitochondrial biology (e.g., mitochondrial fitness/integrity/function/activity)

Autophagy and…….. in ischemic aged livers

Author Response

Thanks for your comments and apologies for our mistake.

The new title has been changed to "Mitochondrial autophagy in ischemic aged livers"